# PufCB-Auth: A lightweight continuous multi-factor authentication scheme integrated PUF with charging behavior features for EV charging

Chongchao Zhang[1,2], Kaichen Zhang[1], Chunguang Zhang[1], Shihui Chen[1], Ji Ma[1,2]*, Shichang Fu[1], Chunlei Yang[1], Yi Wang[3], Nan Zhang[2], Li Zhao[3], Xiaolei Ying[3], Leijie He[3], Hongyue Ma[2]

**1** State Grid Si Ji Wang An Technology Co., Ltd., Beijing, China, **2** State Grid Information & Telecommunication Group Co., Ltd., Beijing, China, **3** Ningbo Power Supply Company of State Grid Zhejiang Electric Power Co., Ltd., Ningbo, Zhejiang, China

* 13641038743@139.com

## Abstract

As electric vehicles (EVs) gain widespread adoption, interactions between EVs and charging infrastructure are increasing, driving the need for secure and efficient authentication methods. However, existing authentication approaches are inadequate to address the unique challenges of dynamic EV charging scenarios. Moreover, they often suffer from static credentials, high computational overhead, and limited adaptability to dynamic user behavior and environmental variability. To address these challenges, this paper proposed PufCB-Auth, a lightweight multi-factor authentication scheme that integrates hardware-level Physical Unclonable Functions (PUFs) with charging behavior features to generate a multi-modal digital fingerprint. To alleviate the negative effects rooting from EV user's charging behavior drift and PUF response fluctuations, the paper also proposed Enhanced PufCB-Auth by incorporating a fingerprint update mechanism. The proposed scheme achieves lightweight design, strong robustness, and continuous authentication capability, making it well-suited for dynamic and resource-constrained EV charging environments. Simulation results validate its effectiveness in improving authentication accuracy and robustness, with minimal system overhead, enabling practical deployment in real-world ChaoJi charging pile–EV interaction environments.

## 1. Introduction

The rapid proliferation of electric vehicles (EVs) and the widespread deployment of charging infrastructure have significantly increased the frequency and complexity of charging pile–EV interactions. According to the report of International Energy Agency

**Data availability statement:** All relevant data are within the paper and its Supporting Information files.

**Funding:** The author(s) received no specific funding for this work.

**Competing interests:** The authors have declared that no competing interests exist.

(IEA), the number of EVs has exceeded 45 million in 2023, with 14 million new EVs sold in this year, accounting for 20% of the global automotive market [1]. Meanwhile, the number of public charging piles worldwide has surpassed 5 million by the end of 2024, with China representing 20% of global automotive sales [2].

This large-scale growth amplifies security risks in unauthenticated charging pile–EV interactions, such as unauthorized access to charging infrastructure and data leakage. However, the traditional security methods applied in charging pile–EV authentication are inadequate for this evolving landscape. Firstly, static password-based approaches and biometric authentication schemes (e.g., fingerprint recognition) exhibit limited resilience against spoofing and physical tampering, and are particularly vulnerable to modern counterfeiting techniques such as 3D printing [3], Secondly, most existing methods only focus on initial authentication, lacking continuous identity verification capabilities. This leaves the system susceptible to per-sistent threats such as identity theft, behavioral impersonation, and hardware cloning. In addition, traditional schemes are mostly based on software encryption [4], which is difficult to resist physical hardware tampering or cloning attacks. Moreover, mali-cious attackers may exploit multiple attack vectors, including impersonating legitimate users, extracting sensitive identity or location data [5], and stealing secret parameters from the device memory through physical attacks. These challenges are exacerbated by the resource-constrained nature of many EV and charging station devices, which renders most cryptographic solutions with high performance impractical [6].

In particular, the rapid deployment of high-power charging technologies, such as the ChaoJi standard [7], introduces new dimensions of risk. Unlike conventional charging, the system supports ultra-high-power DC charging up to 900 kW and 600 A, enabling rapid energy transfer over short durations. However, such high current transmission significantly elevates the risk of thermal stress, equipment damage, or even grid instability in the event of authentication failure or malicious access. For example, malicious access to high-power interfaces could result in equipment over-load, thermal damage, or even grid-level disturbances, especially if attackers manip-ulate the charging identity or session parameters. Under such conditions, faults or attacks are no longer isolated to individual vehicles or piles but may propagate to the wider power infrastructure. This makes it imperative to develop lightweight, robust, and adaptive authentication schemes that are specifically tailored for next-generation charging pile–EV scenarios, including ChaoJi-enabled ultra-fast charging.

Therefore, this paper proposes a lightweight multi-factor continuous authentication scheme for charging pile–EV interaction, which utilizes the PUF features which are difficult to replicate, and combines the EV charging behavior factors as an authenti-cation complement to enhance its resistance in the face of physical noise. Through this approach, the performance of the authentication scheme is comprehensively improved. In addition, the feature update method is added to make the authentication process more adaptable.

The contributions of the research can be summarized in the following 3 aspects:

1.   Multi-Factor Authentication Framework: We propose PufCB-Auth, a collaborative framework combining PUF-based hardware authentication and charging behavior

analytics, tailored for high-power ChaoJi charging pile–EV interaction scenarios. Experimental results validate the independent effectiveness of each factor, demonstrating their capability to resist physical tampering, spoofing attacks, and behavioral anomalies.

2, Lightweight Digital Fingerprinting: A unified authentication identity is created by encoding PUF responses and behavior patterns into a compact digital fingerprint. The design reduces computational and communication overhead, enabling real-time deployment in resource-constrained EV and charging pile systems.

3, Dynamic Feature Update Mechanism: A time-triggered, behavior-deviation-aware update scheme is designed to refresh authentication features in response to PUF drift or environmental changes. This ensures the framework remains resilient to long-term environmental and behavioral variations while maintaining real-time authenticity.

## 2. Related work

### 2.1. PUF-based authentication

Physical Unclonable Functions (PUFs) leverage intrinsic micro-physical variations introduced during the integrated circuit manufacturing process to generate unique, non-replicable responses, making them suitable for device authentication and key generation [8]. Compared to conventional key storage methods, PUFs offer inherent advantages in resisting physical attacks and hardware tampering, which has led to their widespread adoption in resource-constrained environments such as Internet of Things (IoT) devices and edge computing nodes. Yang [9] proposed two lightweight bidirectional authentication protocols based on PUFs and multiple trusted authorities. These protocols employ fuzzy extractors to mitigate noise interference in PUF responses and leverage the lightweight computational characteristics of PUFs to reduce both communication and computation overhead. Additionally, the protocols address camouflage attacks by utilizing the unclonable nature of PUFs. Shlomi et al [10] introduced a vehicle-to-vehicle authentication scheme using instantaneous optical responses in conjunction with radio-based key exchanges. These optical signals are non-forwardable and are derived from certified PUF-based devices embedded in the front and rear of vehicles, enhancing the integrity of inter-vehicle communication.

Despite their advantages, PUFs have known limitations. Machine learning has the potential to predict the response to random challenges through modelling [11]. Gebali [12] mentioned in his review that the reliability of PUFs is affected by environmental factors, and response inconsistency is also one of its main limitations. For this reason, Modarres [13] proposed an efficient lightweight authentication protocol based on noisy PUFs, which combines the properties of noisy PUFs as well as conventional PUFs to address the vulnerability of PUFs to environmental noise and machine learning attacks. However, it is difficult to implement their use of the two types of PUFs to authentication scenarios in electric vehicles.

In summary, while PUF-based authentication provides strong hardware-level protection, it lacks adaptability in the face of behavioral dynamics and environmental variability—highlighting the need for complementary authentication modalities.

### 2.2. Behavior-based authentication for EV charging service

In recent years, some researchers have also investigated how to utilize user charging behavior for authentication through machine learning and anomaly detection techniques. Kern et al. [14] proposed a hybrid intrusion detection system that combines regression-based prediction with anomaly detection to identify potential security threats during EV charging sessions. By incorporating fine-grained behavioral patterns, the system significantly enhances attack detection accuracy. In 2020, Li and Wang [15] developed a machine learning based algorithm to predict EV users' dwell time and energy consumption. They proposed the Ensemble Predicting Algorithm (EPA), which improves the performance and reduces the prediction error by 11%. Yu et al [16] used introducing travel chains and logistic regression model to identify the factors that significantly influence charging behavior, and built a prediction model for EV charging behavior based on single and multiple significant influencing factors.

Although behavior feature-based authentication technology shows great potential in EV charging systems, its practical application still faces many challenges, as shown by [15] study that user charging habits may change due to daily changes, new charging stations, or vehicle upgrades, necessitating adaptive systems to maintain accuracy, suggesting that behavior-based methods alone may be insufficient for robust identity verification

### 2.3. Multi-factor authentication schemes

Based on the above studies, it is evident that single-factor authentication schemes possess inherent limitations. Once the single authentication factor is compromised—whether through spoofing, physical tampering, or behavioral mimicry—the entire authentication process becomes susceptible to failure. Moreover, no existing authentication factor offers absolute security under all conditions.

To address these challenges, researchers have explored multi-factor authentication (MFA) approaches that combine multiple identity dimensions to enhance robustness. Sadri [17] proposed a new two-factor authentication protocol for IoV (Internet of Vehicle), which addressed the shortcomings of IoV authentication that is susceptible to attacks such as sensor capture attacks, user traceability attacks, etc. Al Sibahee et al. [18] combined PUF-based identities, random numbers, and user passwords in their two-factor protocol. This scheme was formally verified under the real-or-random (RoR) model, demonstrating provable security. Subran [19] incorporated human biometrics (particularly fingerprint recognition) into their multi-factor authentication framework to prevent identity theft, man-in-the-middle, and brute-force attacks. The experiments validated the robustness of the proposed scheme compared to traditional single-factor methods.

However, multi-factor authentication often has higher security performance than single factor, however, its authentication scheme may be more cumbersome, which poses a great challenge to the resource-constrained charging pile–EV interaction environment. Therefore, a key research objective is how to design lightweight multi-factor authentication protocols that can be efficiently deployed in such constrained settings without compromising security.

### 2.4. Digital fingerprint–based methods

Digital fingerprint (DF)–based authentication has recently gained increasing attention in the field of identity verification. This technique generates unique and hard-to-forge digital identifiers by integrating intrinsic hardware characteristics and user-specific behavioral features at both the device and user levels. Compared with multi-factor authentication, DF-based techniques offer lightweight alternatives by enabling identity authentication through the comparison of pre-generated digital identifiers, thereby significantly reducing computational overhead.

Among the existing studies on digital fingerprints, Eckersley [20] explored the uniqueness of browser fingerprints and pointed out that they can be used to identify devices, but there is a risk that they can be circumvented by users who change their settings or use a new device. Stylios [21] further extended DF technique by incorporating behavioral features, and proposed a continuous authentication scheme that verifies user identity during device interaction. While this scheme improves both security and user experience compared to static authentication schemes, it still faces challenges due to the dynamic nature of user behavior, which may evolve over time or be imitated by sophisticated attackers. Addressing this issue requires the incorporation of stable hardware features and adaptive update mechanisms that ensure robustness against dynamic user.

### 2.5. Limitations of existing approaches and design motivation

While existing authentication techniques each offer certain advantages, they also present notable limitations when applied independently. Single-factor methods—whether based on hardware properties or user behavior—remain vulnerable to spoofing, physical tampering, or behavioral mimicry. Moreover, no single authentication modality provides sufficient robustness across diverse and dynamic operating conditions commonly found in charging pile–EV interaction scenarios.Multi-factor

authentication (MFA) schemes attempt to address these shortcomings by combining multiple identity factors. Although MFA improves security, it often introduces high computational and communication overhead, making it difficult to implement efficiently on resource-constrained charging devices. Similarly, digital fingerprinting offers a lightweight alternative by encoding device and behavioral features into a unified identifier. However, existing DF-based schemes are typically static and rely heavily on user behavior, which may drift over time or be imitated by adaptive adversaries.

These limitations underscore the need for an authentication framework that is not only secure and lightweight but also adaptable to dynamic physical and behavioral conditions. To this end, we propose a novel authentication scheme that integrates the physical uniqueness of PUFs with the temporal variability of EV charging behavior to construct a multi-modal digital fingerprint. A dynamic fingerprint update mechanism is further introduced to maintain long-term authentication validity in the presence of behavior drift and environmental noise. The scheme is tailored for emerging high-power charging scenarios, such as those based on the ChaoJi charging standard [7]. In such contexts, authentication breaches may threaten not only user safety but also the stability of the power infrastructure.

Overall, the pro-posed approach offers a lightweight, scalable, and secure authentication solution de-signed to meet the unique demands of next-generation charging pile–EV interactions in ultra-fast charging environments.

## 3. PufCB-auth: A lightweight multi-factor continuous authentication scheme integrated PUF with charging behavior features

This section presents PufCB-Auth, a lightweight and adaptive multi-factor authentication scheme designed for charging pile–EV interaction scenarios—particularly under high-power, real-time conditions such as those defined by the China ChaoJi charging standard [7]. The scheme addresses both device-level and user-level security requirements by combining two complementary authentication factors: the physical uniqueness of the EV hardware (captured through PUFs), and the temporal behavioral patterns of EV charging activities.

Fig 1 illustrates the overall architecture. The authentication process begins with a challenge-response scheme based on PUF to verify the physical identity of the EV device. In parallel, the system monitors and evaluates real-time charging behavior to detect anomalies or unauthorized usage patterns. These two factors are then integrated into a dynamic digital fingerprint, which serves as a unified, compact representation of the authentication subject. To maintain long-term

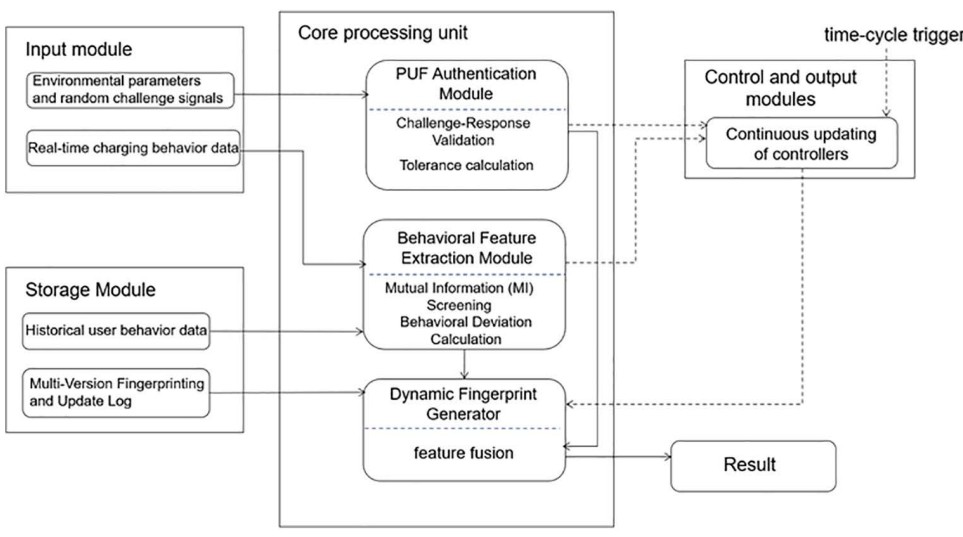

**Fig 1. The Overall Framework of PufCB-Auth based on Multi-factor Integration.**

robustness and adaptability, a feature update method dynamically refreshes the fingerprint in response to environmental changes, user behavior drift, or time-based policies.

The remainder of this section is organized as follows. Section 3.1 details the PUF-based authentication process for capturing hardware-level identity. Section 3.2 describes the extraction and modeling of EV charging behavior features for anomaly detection. Section 3.3 presents the fusion strategy that integrates both factors into a digital fingerprint for efficient, real-time authentication. Section 3.4 explains the dynamic feature update mechanism designed to ensure the long-term validity and resilience of the fingerprint under changing behavioral and environmental conditions.

### 3.1. Puf-auth: A device-level EV-pile authentication based on hardware PUF challenge-response mode

This subsection proposes Puf-Auth, a device-level authentication module based on a hardware PUF-enabled challenge–response protocol. The goal is to establish a unique and tamper-resistant identity for each EV device by leveraging physical randomness embedded in EV's Battery Management System (BMS). During the registration phase before authentication, the charging pile's master control unit sends a randomly generated challenge sequence C to EV, which length is n.

$$C = \{c_1, c_2, \ldots, c_n\}, \ c_i \in \{0, 1\} \tag{1}$$

The EV generates a unique response signal Rref based on the unique physical structure of its BMS system as a unique physical identification of EV, and the process of challenge generation and response is shown on the left side of Fig 2.

$$R_{ref} = PUF(C) = \{r_1, r_2, \ldots, r_m\}, r_i \in \{0, 1\} \tag{2}$$

During real-time authentication, as shown on the right side of Fig 2, the charging pile's master control unit generates a new random challenge sequence, Ctest, and sends it to EV. Once it receives the challenge, it generates a new response sequence Rtest, and returns it to the pile. The pile compares the real-time response Rtest with the baseline response Rref stored during registration, and derives the authentication result by calculating the similarity between the two responses. The similarity is calculated as follows.

$$Sim\left(R_{test}, R_{ref}\right) = \frac{\sum_{i=1}^{m} \delta\left(r_{test,i}, r_{ref,i}\right)}{m} \tag{3}$$

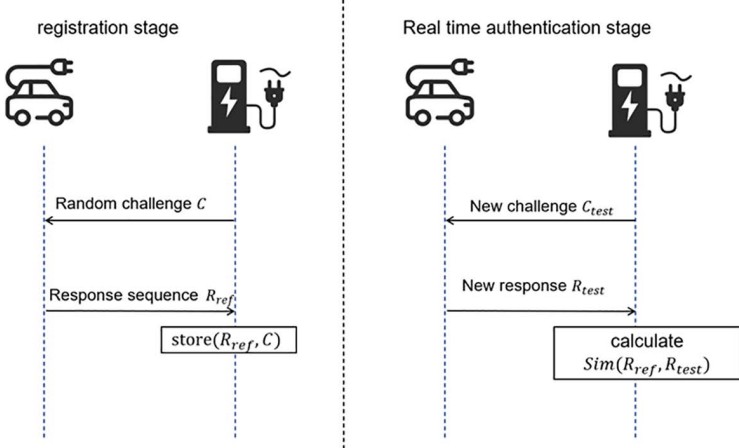

**Fig 2. The Messages Exchange for Puf-Auth.**

where δ(x,y) is the bit comparison function:

$$\delta(x, y) = \begin{cases} 1, & x = y \\ 0, & x \neq y \end{cases} \tag{4}$$

Meanwhile, since the PUF response is sensitive to charging environmental factors, a certain response tolerance is set to ensure that legitimate devices can still pass the authentication under natural fluctuations. Then the equation for authentication is defined as follows.

$$\text{Sim}\left(R_{test}, R_{ref}\right) \geq \tau - \epsilon \tag{5}$$

Where, τ is the set similarity threshold, generally take 0.95, $\epsilon$ is the tolerance range, according to the actual situation to adjust.

In addition, the module incorporates several defense mechanisms to resist common attacks targeting Puf-Auth as follows.

1, to prevent replay attacks, random salt values Salt ∈ (0, 1) n are embedded in the challenge sequence to generate extended challenges, ensuring that each challenge is unique and unpredictable.

2, to counter machine learning–based modeling attacks, a dynamic challenge rotation strategy is used to avoid the attacker from building a PUF mathematical model through finite challenge-response pairs. After each authentication, a new challenge is randomly selected from the pre-generated challenge pool.

3, to hinder statistical analysis and increase unpredictability, each response bit has a small probability $p$ of being randomly flipped during generation. This lightweight obfuscation mechanism reduces response regularity and increases the difficulty of modeling attacks.

### 3.2. CB-auth: A user-level EV-pile authentication based on EV charging behavior features

This subsection proposes CB-Auth, an adaptive and dynamic authentication module based on electric vehicle (EV) charging behavior. The module begins by extracting key behavioral features from historical charging data and representing them as a feature vector B, which serves as the behavioral baseline of the EV user. It consists of statistical representations (i.e., frequency, average value, and ratio) of key behavioral attributes across spatio-temporal patterns, charging demand, and interaction dynamics. The values of each of these dimensions can be obtained by aggregating EV's historical charging data, and the specific feature extraction method is described in Appendix A in S1 File.

$$B = \left[t, L, Q, f, \Phi_{ratio}\right] \tag{6}$$

The sliding window technique is used to save the most recent S charging records and update the behavior baseline vector in real time. Let the spatio-temporal features of the EV user during this period be $t_j$, $L_j$, the charging demand features be $Q_j$, $fj$, and the interaction dynamics features $\Phi_{ratio,j}$. The current behavioral baseline vector, denoted as $B_{base}$, is constructed by aggregating these features across the window. $B_{base}$ is continuously updated when the sliding window slides forward to ensure adaptability to recent behavioral patterns.

$$B_{base} = \left[t_j, L_j, Q_j, f_j, \Phi_{ratio,j}\right] \tag{7}$$

During the charging process, the charging pile obtains the EV user's current charging behavior features Btest in real time:

$$B_{test} = \left[t_{test}, L_{test}, Q_{test}, f_{test}, \Phi_{ratio,test}\right] \tag{8}$$

The degree of behavioral consistency is calculated by finding the probability value of the current behavior in the historical distribution. A low probability indicates a deviation from typical behavioral patterns and may suggest a potential anomaly. To quantify this deviation, the negative logarithm of the probability is introduced as an anomaly score. If the probability of the current behavior $x$ is high, the anomaly score $\Delta_x$ takes a small value, and vice versa, $\Delta_x$ takes a large value.

Based on the importance of different features, weights $\omega_x$ are assigned to each feature to calculate the weighted deviation score $\Delta$:

$$\Delta = \omega_t \cdot \Delta_T + \omega_L \cdot \Delta_L + \omega_Q \cdot \Delta_Q + \omega_f \cdot \Delta_f \tag{9}$$

Where $\omega_t + \omega_L + \omega_Q + \omega_f = 1$. A larger value of $\Delta$ indicates a higher risk of abnormal behavior. If $\Delta$ is close to 0, it indicates that the current behavior highly matches the historical pattern.

Subsequently, based on the set of historical anomaly scores from the user's most recent $S$ charging sessions, denoted as $\{\Delta_1, \Delta_2, ..., \Delta_S\}$, its mean $\mu_\Delta$ and standard deviation $\sigma_\Delta$ are calculated. The proposed behavior-based authentication does not assume that an EV user's raw charging behavior features (e.g., time, location, or demand) follow a normal distribution. Each user may exhibit distinct and potentially multi-modal behavior patterns, which constitute the inherent behavioral profile on which authentication is based. The dynamic threshold is defined as $\tau_\Delta$:

$$\tau_\Delta = \mu_\Delta + 3\sigma_\Delta \tag{10}$$

For each new charging record, $\mu_\Delta$ and $\sigma_\Delta$ are recalculated to ensure that the thresholds are adapted to the user's latest behavior patterns. The authentication decision relies on a deviation score that quantifies how far a current charging event departs from the user's own established behavioral profile. Thresholds are defined relative to the dispersion of this deviation score for each individual user, allowing the authentication process to adapt to different levels of behavioral variability across users. Based on the comparison results of the composite score $\Delta$ and the dynamic threshold $\tau_\Delta$, different levels of security responses are triggered. If $\Delta \leq 0.3\tau_\Delta$, it is regarded as low risk and no additional authentication is required. In the actual data, 0.3 is a good threshold under which almost all attackers' forged data can be excluded. If $0.3\tau_\Delta < \Delta \leq \tau_\Delta$, it is regarded as medium risk and triggers a light secondary validation. If $\Delta > \tau_\Delta$, it means this authentication deviation is beyond $3\sigma$, which is statistically a small probability event. The behavior data of normal users almost never appear, so it is treated as high risk. Immediately block the charging request, force the PUF hardware to re-authenticate, and trigger the digital fingerprint update process.

Non-parametric approaches such as percentile- or IQR-based thresholds were not adopted because they inherently label a fixed proportion of observations as anomalous, even when all behaviors remain close to the user's typical pattern. In contrast, the proposed variance-adaptive thresholding strategy permits all observations to be treated as low risk when user behavior is stable, which better aligns.

### 3.3. PufCB-auth: A multi-factor EV-pile authentication based on digital fingerprint integrated PUF with charging behavior features

Building upon the two-factor authentication integrated PUF responses with user charging behavior features, this subsection proposes a unified digital fingerprinting scheme, referred to as PufCB-Auth. This scheme supports rapid, lightweight authentication and is well-suited for resource-constrained EV-pile interaction environments, while also providing adaptability to physical and behavioral dynamics.

The digital fingerprint is generated by the baseline integration of PUF physical features and user charging behavior features. Firstly, the recent S historical PUF responses are statistically analyzed, and the mean value of the PUF responses, Ravg, is computed as the physical features of the PUF module of the on-board battery:

$$R_{avg} = \frac{1}{S} \sum_{i=1}^{S} R_i \tag{11}$$

The trusted hardware identifier DPUF is generated by hashing the average of recent S PUF responses $R_{avg}$, as follows:

$$D_{PUF} = Hash\left(R_{avg}\right) \tag{12}$$

Based on the sliding window technique, the EV user's spatio-temporal features are extracted from the most recent S charging records as in Eq. (7) to form the current behavior baseline vector B. The unique fingerprint df is generated by concatenating the PUF-derived identifier DPUF with the behavioral baseline Bbase and then hashing the combined vector as follows.

$$df = Hash(D_{PUF}||B_{base}) \tag{13}$$

## 3.4. Enhanced PufCB-auth based on continuous digital fingerprint update

In order to avoid the digital fingerprint to be invalidated due to changes in hardware and behavior features, we propose an enhanced version of PufCB-Auth through a continuous digital fingerprint update mechanism triggered by the following three conditions.

Time update condition: mandatory update every fixed period $T$ (e.g., 30 days).

Anomaly detection update condition: deviation score Δ exceeds the threshold $\tau_\Delta$, or the system detects a high-frequency alarm.

PUF response drift update condition: the PUF response similarity decreases less than the threshold value due to environmental disturbances.

When the respective update condition is triggered, the system generates a new fingerprint and replaces the old one according to the following steps:

At the hardware layer, the system counts the most recent $S$ PUF responses and computes the statistical mean value of the new PUF response, $R_{avg\_new}$, as the updated physical characteristic of the device, to generate the new identification $D_{PUF-new}$ and update the behavior baseline $B_{new}$ from the most recent $S$ charging records

Compute the new PUF identifier with the updated behavior baseline and hash to get the new digital fingerprint:

$$df_{new} = Hash(D_{PUF-new}||B_{new}) \tag{14}$$

To prevent data leakage or tampering, the system uses symmetric encryption to protect the new fingerprint and replace the old fingerprint, and subsequently record the update event in the security log to achieve traceability and non-tampering.

To prevent update poisoning and denial-of-service (DoS) attacks, the proposed fingerprint update mechanism is designed with strict admission and rate-control policies. For behavior-based authentication, newly observed high-risk charging behaviors do not directly trigger profile updates. Only historical charging records that have previously passed authentication are eligible for inclusion in the update dataset, while high-risk observations are used exclusively for risk assessment. This design prevents adversarial or abnormal behavior from contaminating the behavior profile, even under repeated attack attempts.

To accommodate legitimate long-term behavior evolution, profile updates are rate-limited and executed only at controlled frequencies. When update triggers occur excessively, the system escalates to an auxiliary verification stage requiring explicit user confirmation. If verification fails, the authentication process is locked and no update is performed; only after successful confirmation are corresponding records admitted into the update dataset.

For hardware-derived identity factors, such as analog PUF responses, physical characteristics are expected to evolve gradually due to aging or environmental variation. Accordingly, physical feature updates are performed conservatively using only recent authentication-passed records. If significant deviation persists even after incorporating recent legitimate

data, the system interprets this as evidence of device replacement or impersonation rather than benign drift, and the device is rejected instead of being adaptively accommodated.

## 4. Experimental design and performance analysis of PufCB-Auth

### 4.1. Evaluation setup

The experimental evaluation is conducted on a controlled simulation and emulation platform designed to reproduce electric vehicle charging and authentication scenarios. Hardware-level identity information is emulated using simulated PUF responses derived from heterogeneous physical channels, following the PUF construction and noise modeling described in Section 3 and Appendix B in S1 File. In our target deployment, the "PUF factor" is instantiated from inherent, device-specific micro-variations of the EV Battery Management System (BMS), which provide challenge–response style uniqueness similar to PUF behavior. Rather than assuming a specific conventional silicon PUF primitive (e.g., SRAM PUF or ring oscillator PUF), we model this factor as a BMS-derived PUF-like hardware fingerprint. This implementation is intended to evaluate the authentication framework and its robustness to noise/drift, rather than to benchmark a particular silicon PUF circuit. The proposed framework is compatible with standard PUF primitives if available in a production BMS/ECU.

Charging behavior data are synthetically generated to simulate long-term user charging patterns under different usage habits and environmental conditions. Behavioral features are extracted from generated charging sessions and evaluated in a continuous authentication setting, where authentication decisions are made throughout the charging process rather than at a single handshake. This unified experimental framework allows systematic evaluation of hardware-based authentication, behavior-based authentication, and their combination under identical conditions.

Two representative lightweight authentication approaches are selected as external baselines for comparison. The first baseline is a PUF-based authentication scheme designed for Internet of Vehicles environments, which relies exclusively on hardware-level challenge–response behavior to establish device identity. This type of scheme represents PUF-only authentication paradigms that emphasize unclonability and low computational overhead but do not account for user behavior dynamics or long-term interaction patterns.

The second baseline is a behavior-driven authentication scheme (L2AI), which performs authentication by modeling charging behavior characteristics and detecting deviations from learned behavior profiles. This approach focuses on identifying abnormal or malicious behavior without incorporating hardware-derived identity information. Together, these two baselines represent hardware-centric and behavior-centric authentication paradigms, respectively, and serve as references for evaluating the security coverage and robustness of different design choices.

### 4.2. Comparison with state-of-the-art lightweight authentication schemes

This section compares the proposed scheme with representative state-of-the-art lightweight authentication schemes from the literature. The comparison focuses on efficiency-related metrics and security capability coverage, as defined in Section 4.3.

Specifically, we first introduce a unified adversary model and security evaluation methodology to ensure that different schemes are assessed under comparable assumptions. We then compare communication, computational, and storage overheads to evaluate deployment feasibility on constrained devices. Finally, security performance is analyzed through a detailed comparison of attack resistance and adversarial capability requirements, with particular attention to long-term impersonation and behavior mimicry threats that are not fully addressed by conventional one-time authentication schemes.By structuring the comparison in this manner, this section aims to provide a fair, transparent, and reproducible assessment of how different lightweight authentication designs balance security coverage and implementation efficiency, while explicitly avoiding misleading conclusions drawn from incompatible evaluation metrics.

**4.2.1. Threat model and security evaluation methodology.** To enable a fair and systematic security comparison across different authentication schemes, this subsection first defines a unified threat model and then describes the methodology used for security evaluation.

Threat model. We consider a probabilistic polynomial-time adversary A operating in an electric vehicle (EV) charging environment, with capabilities commonly assumed in authentication protocol analysis for IoT and vehicular systems. Specifically, the adversary is assumed to have the following abilities:

(C1) Network control: A can eavesdrop, replay, delay, and modify messages transmitted over the public communication channel between the EV-side unit and the charging infrastructure, following a Dolev–Yao–style network adversary model.

(C2) Impersonation attempts: A may attempt to impersonate a legitimate EV to a charging pile or impersonate a legitimate charging pile to an EV, with the goal of unauthorized charging access or service misuse.

(C3) Partial device compromise: A may obtain partial information from a compromised device, such as non-volatile stored parameters or temporary authentication states. However, A is assumed to be unable to perfectly clone the intrinsic physical behavior of genuine PUF hardware or reconstruct its exact challenge–response behavior.

(C4) Behavioral manipulation and observation: A may observe long-term charging behavior patterns and attempt to mimic or manipulate charging behavior records, including injecting abnormal or crafted charging events to deceive behavior-based authentication components.

(C5) Denial-of-service and update abuse: A may attempt to degrade system availability by triggering repeated authentication failures or maliciously inducing frequent profile updates.

The threat model does not assume that the adversary can break standard cryptographic primitives (e.g., hash functions) or fully compromise all authentication factors simultaneously without detection.

Security evaluation methodology. Under the above threat model, security performance is evaluated using a structured, qualitative comparison methodology. For each authentication scheme, we analyze its resistance to a standard set of attacks, including replay attacks, man-in-the-middle attacks, impersonation attacks, partial device capture, behavior mimicry, privacy leakage, and long-term masquerading.

Instead of claiming absolute security guarantees, the evaluation focuses on identifying the adversarial conditions required for a successful compromise. In particular, we compare (i) which authentication factors must be forged or compromised, (ii) whether a single compromised factor is sufficient to break authentication, and (iii) whether the scheme can sustain security over long-term interactions beyond the initial authentication phase.This threat-model-driven analysis enables a consistent comparison between one-time authentication schemes and the proposed continuous, multi-factor authentication framework, while avoiding misleading conclusions drawn from incompatible evaluation metrics or assumptions.

**4.2.2. Security analysis and attack coverage comparison.** Based on the unified threat model defined in Section 5.4.1, this subsection analyzes how different authentication schemes respond to representative attack behaviors that naturally arise under the assumed adversarial capabilities (C1–C5). For each attack category, we first explain why the attack is feasible under the threat model, and then examine the protection mechanisms and limitations of each scheme.

1) Replay attacks: Under capability C1, the adversary can eavesdrop and record authentication messages exchanged over the public channel, making replay attacks a fundamental threat in open charging environments. The PUF-MAP scheme mitigates replay attacks by binding authentication tokens to fresh challenges and nonces, ensuring that previously observed responses cannot be reused in a different session. Similarly, L2AI employs timestamps and random numbers in its authentication messages, which invalidate replayed messages outside their validity window. The proposed PufCB-Auth scheme inherits this protection by using session-specific PUF challenges and freshness parameters. Therefore, simple replay of recorded messages is insufficient to pass authentication in all considered schemes.

2) MITM attacks: MITM attacks also stem from the network control capability (C1), where the adversary attempts to intercept and manipulate messages during authentication. In the PUF-MAP scheme, MITM resistance is achieved by cryptographically binding authentication messages to unpredictable PUF challenge–response pairs, preventing an adversary from modifying messages without invalidating verification. L2AI similarly relies on one-way hash computations and shared secrets to ensure message integrity during authentication exchanges. PufCB-Auth provides equivalent session-level protection against MITM attacks. In addition, because authentication decisions depend on both hardware-derived PUF responses and behavior-derived fingerprints, message manipulation alone cannot enable sustained authentication unless the adversary also satisfies the independent identity factors.

3) Impersonation Attacks: Impersonation attacks arise from capability C2, where the adversary attempts to masquerade as a legitimate EV or charging pile. In PUF-MAP schemes, successful impersonation requires the adversary to accurately reproduce or model the target device's PUF responses. As long as the PUF remains unclonable and secret, impersonation is considered infeasible. In L2AI, impersonation resistance is primarily based on protecting credentials and biometric-related information. Once these factors are compromised, the scheme does not introduce additional independent barriers to prevent impersonation. In contrast, PufCB-Auth increases the impersonation threshold by requiring both correct PUF behavior and consistency in long-term charging behavior. Even if one factor is compromised, impersonation cannot be sustained without simultaneously satisfying the other, independent factor.

4) Partial device capture: Under capability C3, the adversary may extract partial information from a compromised device, such as stored parameters, helper data, or transient authentication states, without fully controlling the device or replicating its physical characteristics. In PUF-MAP schemes, the impact of partial device capture depends on the scope of leaked information. Although secret keys are not explicitly stored, auxiliary data and protocol states may still be exposed, which can reduce the security margin and facilitate repeated authentication attempts under certain conditions. In L2AI, authentication relies on stored credentials and biohash-related parameters. Once such information is partially leaked, a single compromised factor may be sufficient to enable impersonation, as no independent hardware-rooted identity is required. In contrast, PufCB-Auth mitigates partial device capture by eliminating single points of failure. Even if stored behavior-related parameters are exposed, successful authentication still requires uncontaminated PUF responses and sustained behavior consistency, which cannot be reproduced solely from leaked data. As a result, the scheme provides stronger resilience against partial device compromise.

5) Behavior mimicry and long-term masquerading: Capability C4 enables the adversary to observe and manipulate long-term charging behavior, leading to behavior mimicry and long-term masquerading attacks.Neither PUF-MAP schemes nor L2AI explicitly address behavior mimicry, as authentication is completed during session initiation and does not consider post-authentication behavior. Consequently, once initial authentication is passed, these schemes do not detect sustained masquerading attempts. PufCB-Auth explicitly targets this threat by introducing continuous authentication throughout the charging process. By periodically validating behavior consistency in conjunction with hardware identity, the scheme limits the adversary's ability to maintain long-term masquerade through short-term imitation or injected records.

6) Privacy leakage: Under capabilities C1 and C4, adversaries may infer sensitive information by observing authentication messages or behavior-related data. PUF-MAP schemes typically transmit identity-related tokens and challenge–response information, which may enable limited inference if identifiers are reused. L2AI employs pseudo-identities but may still expose behavioral or contextual attributes during authorization. PufCB-Auth reduces privacy leakage by transmitting only compact, non-invertible behavior descriptors and pseudonymous identifiers, rather than raw charging trajectories or precise location data, thereby limiting information exposure under passive observation.

Table 1 summarizes the resistance of the considered schemes to representative attack categories under the unified threat model.

## 4.3. Efficiency analysis and overhead comparison

This section compares the proposed PufCB-Auth scheme with L2AI and PUF-MAP, in terms of communication overhead and computational cost. The comparison focuses on the authentication phase, which is the most time-critical stage in real-time EV charging scenarios.

**4.3.1. Communication overhead.** Communication overhead is evaluated by the number of message exchanges and the total transmitted data size during one authentication session.

PufCB-Auth adopts a lightweight PUF-based challenge–response mechanism and performs behavior authentication locally, requiring only two message rounds (challenge and response). PUF-MAP involves interactions among vehicle, RSU, and branch TA, resulting in three to four communication rounds. L2AI requires user–gateway–server–blockchain interactions, leading to at least four rounds, sometimes more due to authorization token validation. Let:

$|C|$ denote the PUF challenge length (128 bits),

$|R|$ denote the PUF response length (128 bits),

$|H|$ denote a hash output (160 bits).

Based on the protocol descriptions:

PufCB-Auth transmits only $|C|+|R|=256$ bits per session. PUF-MAP additionally transmits helper data, identities, and hash values, totaling approximately 640–768 bits. L2AI exchanges multiple hashes, Biohash outputs, authorization tokens, and blockchain indices, resulting in more than 1,000 bits per authentication session.

As shown in Table 2, PufCB-Auth reduces communication overhead by 60–75% compared with L2AI, which is critical for ultra-fast ChaoJi charging scenarios where authentication latency directly affects system safety and efficiency.

**4.3.2. Computational overhead.** Computational cost is evaluated using the number of cryptographic and lightweight operations executed during authentication. Following common practice, the cost is expressed as the number of basic operations:

$T_h$: one-way hash

$T_{xor}$: XOR operation

$T_{PUF}$: PUF response generation

$T_{FE}$: fuzzy extractor operation

$T_{BC}$: blockchain-related processing

**Table 1. Summary of Security Analysis and Attack Coverage Comparison.**

| Attack Type | PUF-MAP | L2AI | PufCB-Auth |
|---|---|---|---|
| Replay (C1) | ✓ | ✓ | ✓ |
| MITM (C1) | ✓ | ✓ | ✓ |
| Impersonation (C2) | ✓ | ✓ | ✓ |
| Partial device capture (C3) | ○ | ✗ | ✓ |
| Behavior mimicry (C4) | ✗ | ✗ | ✓ |
| Long-term masquerade (C4) | ✗ | ✗ | ✓ |

**Table 2. Communication overhead comparison.**

| Scheme | Communication Rounds | Transmitted Data (bits/session) |
|---|---|---|
| PufCB-Auth | 2 | 256 |
| PUF-MAP | 3–4 | 640–768 |
| L2AI | ≥4 | ≥1024 |

Based on protocol workflows, PufCB-Auth requires $1 \times T_{PUF}$, $1 \times$ hamming distance computation and lightweight probability and weighted-sum calculation; PUF-MAP requires $1 \times T_{PUF}$, $1 \times T_{FE}$, 3 or $4 \times T_h$, and multiple XOR operations; L2AI requires: $6–8 \times T_h$, biohash computation, token generation and verification, and blockchain read/write operations. Table 3 shows the comparison of the computing overhead of the three protocols.

PufCB-Auth avoids all heavy cryptographic primitives and centralized processing, achieving the lowest computational complexity among the three schemes.

**4.3.3. Storage overhead.** This subsection evaluates the storage requirements of PufCB-Auth, L2AI, and PUF-MAP by analyzing the long-term security parameters stored at participating entities.

Following common practice, only persistent parameters required for authentication are considered, while ephemeral session variables are excluded. The storage cost is measured in bits and includes: identity-related parameters, cryptographic keys or helper data, authentication tokens or indices. Let:

$|ID| = 128$ bits,

$|H| = 160$ bits (hash output),

$|K| = 128$ bits (key or PUF response),

$|P| = 128$ bits (helper data).

PufCB-Auth should store PUF hardware and sliding window behavior statistics on EV side, store reference PUF response $R_{ref}$ (128bits) and behavior fingerprint template on charging pile side. Total persistent storage per EV–pile pair is approximately: $S_{PufCB-Auth} \approx 128 + 128 = 256$ bits; PUF-MAP should store helper data for fuzzy extractor (PPP, 128 bits), identity and hash-related parameters on vehicle side, store vehicle ID, PUF-derived key, hash values and authentication records on RSU/ TA side, The total storage requirement per vehicle is approximately: $S_{PUF-MAP} \approx | ID | + | K | + | P | + | H | = 128 + 128 + 128 + 160 = 544$ bits. L2AI should store pseudo-identity, biohash output, authorization token, Hash-derived secret values on user device/ smart card, store access control records, token indices, user authentication metadata on server/ blockchain. The total storage requirement per user is at least: $S_{L2AI} \geq 3 | H | + | Biohash | + | Token | \approx 3 \times 160 + 128 + 128 = 736$ bits. In practice, the blockchain-related metadata further increases the storage overhead.

As shown in Table 4, PufCB-Auth reduces storage overhead by ≈53% compared with PUF-MAP and by ≈65% compared with L2AI, making it more suitable for embedded EV and charging pile devices with limited memory resources.

**Table 3. Computational overhead comparison.**

| Scheme | Hash operations | PUF evaluations | Other operations | Estimated computation time (ms) |
|---|---|---|---|---|
| PUF-MAP | 3 | 1 | – | ≈ 0.225 |
| L2AI | 5 | – | Biohash | ≈ 0.310 |
| PufCB-Auth | 3 | 1 | Behavior matching | ≈ 0.275 |

 

**Table 4. Storage overhead comparison.**

| Scheme | EV/ User Storage (bits) | Infrastructure Storage | Centralized Dependency |
|---|---|---|---|
| PufCB-Auth | ≈256 | Minimal | No |
| PUF-MAP | ≈544 | Moderate (TA/RSU) | Yes (multi-TA) |
| L2AI | ≥736 | High (Blockchain) | Yes |

## 4.4. Robustness under noise and behavior drift

This subsection evaluates the robustness of the proposed authentication framework under practical disturbances, including measurement noise and long-term behavior drift. The purpose of this evaluation is to assess the stability and usability of authentication decisions in realistic operating environments, rather than to demonstrate cryptographic security properties.

**4.4.1. Ablation experiment.** This section introduces how to conduct ablation experiments of PUF-Auth and CB-Auth.

To simulate real-world deployment conditions, we construct a hardware prototype that emulates the BMS (Battery Management System) of an EV as described in section 4.1. To test the resistance of PUF features against hardware-level cloning, one Device Under Test (DUT) and five cloned devices—constructed using identical hardware models and components—were used as the experimental setup. The response similarity distributions across all devices are shown in Fig 3. Most of the cloned devices produced responses with similarities ranging from 105/128–116/128 while And the similarity of PUF response keys tested by DUT equipment is 122/128 or above, so we set 0.95 as the authentication pass criteria for each test.

For PUF-Auth, the authentication pass rate was tested under normal environment, temperature fluctuation (−20°C~60°C) and voltage perturbation (±5%) and vibration interference respectively, used the DUT. Each perturbation type was tested 200 times, respectively and 0.95 is token as the authentication threshold.

For CB-Auth, we set S = 100 for the parameter $\tau_\Delta$ to systematically define the behavioral baseline window, which is a relatively appropriate amount of data in the actual operation data. It is enough to summarize the behavior and habits of users in the recent one to three months, and can be adjusted appropriately for different user groups. It is a suitable amount of data in actual operational data which is sufficient to summarize the user's behavior habits in the past one to three months and can be adjusted for different user groups. The uniform standard is used for each EV user's behavior score to ensure consistent evaluation. The scoring is defined as follows.

$$Bscore = \frac{\Delta}{\tau_\Delta} = \frac{\Delta}{\mu_\Delta + 3\sigma_\Delta}$$

(15)

The expriment simulates the following four representative attack types, i.e., Attack 1-Power theft attack, cterized by abnormally high charging frequency and random geographic location jumps; Attack 2-Battery replacement attack, indicated by a mismatch between reported battery characteristics (e.g., charging power) and usual charging locations; Attack 3-Location forgery attack, characterized by static GPS coordinates lacking typical random offsets; Attack 4-Abnormal interruption attack, identified by a significantly increased frequency of session terminations compared to historical user behavior. For evaluation, 200 normal charging records were generated for each of five EV users under consistent parameter settings. In addition, 50 attack samples per attack type (totaling 200 attack samples) were randomly mixed into the EV users' normal data, resulting in 240 records per user (200 normal + 40 attack samples). We passed the certification for low-risk data, but refused to pass the certification for high-risk data. Medium risk data needs to be further judged in combination with the user's recent data change trend.

**4.4.2. PufCB-Auth's authentication performance comparison with single-factor Schemes'.** To verify the effectiveness of our multi-factor fusion authentication scheme (i.e., PufCB-Auth), this experiment compares its

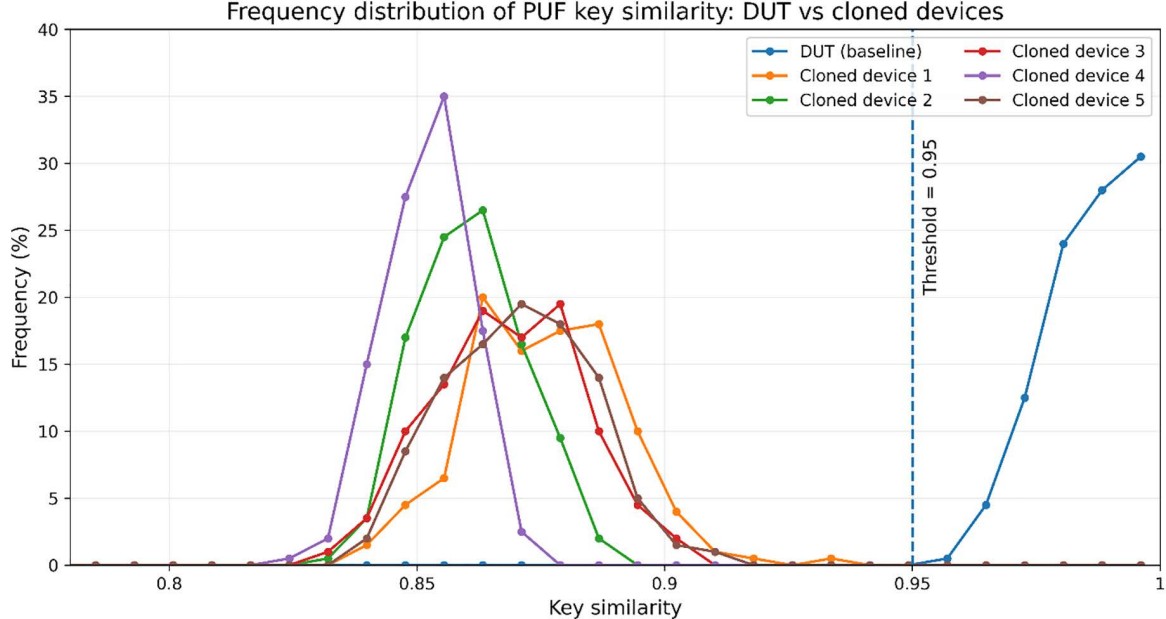

**Fig 3. Statistical Result of Frequency Disbution of Key Similarity for Similar Device Simulation.**

performance against two single-factor baselines, i.e.,PUF-only (Puf-Auth) and behavior-only (CB-Auth) authentication. The goal is to evaluate whether combining hardware-level identity and behavior-based anomaly detection offers measurable advantages in terms of accuracy, robustness, and adaptability, especially under environmental interference and behavioral deviations.

To perform the comparison, we simulate five distinct users, and each assigned a unique combination of a PUF hardware profile and a charging behavior model. For each user, 300 normal charging samples are generated under the premise of unchanged parameters and 200 attack samples (distributed across four typical attack types). In parallel, the three authentication schemes (i.e., PufCB-Auth, Puf-Auth and CB-Auth) are tested under both standard and disturbed environments (e.g., temperature and voltage variations). The digital fingerprint is generated follow the method described in Section 3.3. The authentication succeeds if the fingerprint matches the expected value within a defined tolerance.

Tables 5 and 6 have shown the performance results across the three schemes. Under normal environmental conditions, Puf-Auth maintains nearly 100% user acceptance, but it lacks the capability to detect behavioral anomalies.

**Table 5. Authentication Performance Comparison Between Single-Factor and Multi-Factor Schemes under Normal Environment.**

| Tested users | Single-Factor | | | | Multi-Factor | |
|---|---|---|---|---|---|---|
| | *Puf-Auth* | | *CB-Auth* | | *PufCB-Auth* | |
| | User adoption | Attack detection | User adoption | Attack detection | User adoption | Attack detection |
| A | 100% | N/A | 93.3% | 93.5% | 98.7% | 93.5% |
| B | 100% | N/A | 92.7% | 93.0% | 98.0% | 93.0% |
| C | 99.7% | N/A | 88.0% | 95.0% | 97.3% | 95.0% |
| D | 99.0% | N/A | 95.3% | 90.5% | 98.7% | 90.5% |
| E | 99.3% | N/A | 94.7% | 92.5% | 99.0% | 92.5% |

Table 6. Authentication Performance Comparison Between Single-Factor and Multi-Factor Schemes under Disturbed Environment.

| Tested users | Single-Factor | | | | Multi-Factor | |
| --- | --- | --- | --- | --- | --- | --- |
| | *Puf-Auth* | | *CB-Auth* | | *PufCB-Auth* | |
| | User adoption | Attack detection | User adoption | Attack detection | Attack detection | User adoption |
| A | 92.7% | *N/A* | 93.3% | 93.5% | 97.7% | 93.5% |
| B | 93.3% | *N/A* | 92.7% | 93.0% | 96.0% | 93.0% |
| C | 94.0% | *N/A* | 88.0% | 95.0% | 95.3% | 95.0% |
| D | 89.0% | *N/A* | 95.3% | 90.5% | 96.7% | 90.5% |
| E | 91.0% | *N/A* | 94.7% | 92.5% | 96.0% | 92.5% |

Conversely, CB-Auth shows good attack detection performance (>90%) but is subject to user-dependent variation, for example, for "simulated user C", its authentication performance is significantly lower than that of other individuals. PufCB-Auth achieves a balanced trade-off, with authentication pass rates above 97% and attack detection rates exceeding 93% across all users. This demonstrates that fusing PUF and behavior features not only enhances detection coverage but also mitigates the individual limitations of each single-factor method.

Under disturbed environments, the performance gap becomes more pronounced: Puf-Auth suffers a significant drop in user acceptance due to environmental sensitivity, while CB-Auth maintain relatively stable performance but are constrained by limited generalization. In contrast, PufCB-Auth consistently maintains high performance by leveraging the complementary strengths of both feature types.

**4.4.3. Enhanced PufCB-Auth's performance comparison with single-factor schemes and PufCB-auth before feature update.** This section evaluates the performance gain achieved through Enhanced PufCB-Auth, which is designed to ensure the long-term validity and adaptability of the digital fingerprint as device states and user behaviors evolve.

The digital fingerprint is updated according to the conditions described in Section 3.4. Once meeting the update conditions, the updated features is used to re-authenticate. If the authentication still fails through the updated features, then it is judged as abnormal behavior. To assess the effectiveness of this update mechanism, the test conditions is the same as in Section 4.6.2, but this time compare authentication results before and after performing the feature update. The same five users, 300 normal samples, and 200 attack samples per user are used in both normal and disturbed environments, and the test results are shown in Table 7 and Table 8.

The experimental results demonstrate that the feature update mechanism significantly improves the performance of three schemes, where PufCB-Auth has shown the most consistent and balanced gains. As shown in Table 6, under stable environmental conditions, PufCB-Auth achieves user adoption rates between 97.3% and 100%, and attack detection rates ranging from 91.5% to 96.5% across all five simulated users.

Under disturbed environments, as presented in Table 8, the advantages of the update mechanism become even more pronounced. PufCB-Auth maintains consistently high user adoption rates ranging from 97.0% to 99.3%, along with strong attack detection performance between 91.5% and 96.5%, despite the presence of temperature and voltage fluctuations. In contrast, although Puf-Auth recovers high pass rates through reference profile updates, it cannot still detect behavioral anomalies. Likewise, CB-Auth benefits from behavior model recalibration but still shows sensitivity to individual user variability.

The findings confirm that the feature update mechanism is essential not only for preserving long-term effectiveness but also for enhancing the robustness and adaptability of the PUF-based authentication schemes. In essence, the update mechanism enables continuous identity verification over time and effectively addresses gradual shifts in user behavior and PUF drift.

**Table 7. Authentication Performance Before and After Feature Update in Normal Environment.**

| Tested users | *Puf-Auth* | *CB-Auth* | | *PufCB-Auth* | |
|---|---|---|---|---|---|
| | User adoption | User adoption | Attack detection | User adoption | Attack detection |
| A | 100%→100% | 93.3%→98.3% | 93.5%→95.0% | 98.7%→100% | 93.5%→95.0% |
| B | 100%→100% | 92.7%→98.7% | 93.0%→93.0% | 98.0%→98.0% | 93.0%→93.0% |
| C | 99.7%→99.7% | 88.0%→96.0% | 95.0%→96.5% | 97.3%→97.3% | 95.0%→96.5% |
| D | 99.0%→99.3% | 95.3%→99.0% | 90.5%→91.5% | 98.7%→98.7% | 90.5%→91.5% |
| E | 99.3%→99.7% | 94.7%→97.7% | 92.5%→94.5% | 99.0%→99.0% | 92.5%→94.5% |

**Table 8. Authentication Performance Before and After Feature Update in disturbed environment.**

| Tested users | *Puf-Auth* | *CB-Auth* | | *PufCB-Auth* | |
|---|---|---|---|---|---|
| | User adoption | User adoption | Attack detection | User adoption | Attack detection |
| A | 92.7%→97.3% | 93.3%→98.3% | 93.5%→95.0% | 97.7%→98.7% | 93.5%→95.0% |
| B | 93.3%→97.3% | 92.7%→98.7% | 93.0%→93.0% | 96.0%→99.3% | 93.0%→93.0% |
| C | 94.0%→95.7% | 88.0%→96.0% | 95.0%→96.5% | 95.3%→97.0% | 95.0%→96.5% |
| D | 89.0%→96.0% | 95.3%→99.0% | 90.5%→91.5% | 96.7%→98.3% | 90.5%→91.5% |
| E | 91.0%→97.0% | 94.7%→97.7% | 92.5%→94.5% | 96.0%→98.3% | 92.5%→94.5% |

## 5. Conclusion

The proposed scheme PufCB-Auth, a lightweight and adaptive multi-factor authentication scheme for charging pile–EV interaction scenarios, particularly focuses on emerging high-power charging standards such as ChaoJi. It integrates hardware-level PUFs with dynamic EV charging behavior features to construct a novel digital fingerprint–based continuous authentication scheme. Besides, a feature update mechanism further enhances long-term robustness under changing environments and EV user's charging behaviors. Through extensive simulation and experimental validation, the results have demonstrated that PufCB-Auth has high authentication accuracy and strong attack resistance while maintaining low computational and communication overhead, which makes it well-suited for real-world deployment in constrained charging infrastructure, including ultra-fast charging environments such as those defined by the ChaoJi standard [7].

Despite these promising results, there remain several avenues for future improvement. First, the current thresholds for authentication and anomaly detection are empirically set, which may limit performance optimization under varying conditions. Adaptive or data-driven parameter tuning could further enhance accuracy and flexibility. Second, behavioral feature models exhibit individual variability. The more obvious features of the user attack resistance are stronger, but it is also easier to detect normal behavior as anomalies, while the user with more random features is the opposite, addressing this trade-off between sensitivity and generalization remains a key research direction. Lastly, the feature update scheme, though effective, may be overly sensitive to single anomalies. In order to accurately identify abnormal behavior, the feature update is performed at the first occurrence of abnormal behavior, which results in the model being very sensitive to abnormal behavior, maybe incorporating update frequency control or anomaly accumulation thresholds could improve its robustness.

Overall, the research offers PufCB-Auth, a practical, secure, and lightweight solution for next-generation EV charging environments. It lays a foundation for secure authentication in ChaoJi-enabled ultra-fast charging systems, and contributes to the broader goal of ensuring stability and trust in future smart grid-integrated transportation infrastructures.

## Supporting information

**S1 Dataset. Raw data used for generating simulated EV charging behavior dataset (ZIP).** This file contains the original data used to generate the synthetic user behavior dataset and the corresponding parameter settings for reproducing the statistical analyses and figures reported in this study.
(ZIP)

**S1 File. Appendix.**
(DOCX)

## Author contributions

**Conceptualization:** Chongchao Zhang, Leijie He.

**Data curation:** Shihui Chen, Li Zhao, Xiaolei Ying.

**Formal analysis:** Li Zhao.

**Funding acquisition:** Chongchao Zhang, Ji Ma, Yi Wang, Nan Zhang, Li Zhao, Hongyue Ma.

**Investigation:** Chunguang Zhang, Ji Ma, Chunlei Yang, Yi Wang, Leijie He, Hongyue Ma.

**Methodology:** Chunlei Yang, Yi Wang, Nan Zhang, Xiaolei Ying.

**Project administration:** Chongchao Zhang, Kaichen Zhang.

**Resources:** Nan Zhang, Xiaolei Ying.

**Software:** Shichang Fu, Nan Zhang, Xiaolei Ying.

**Supervision:** Chongchao Zhang, Kaichen Zhang.

**Validation:** Li Zhao, Xiaolei Ying.

**Visualization:** Li Zhao, Xiaolei Ying.

**Writing – original draft:** Ji Ma.

**Writing – review & editing:** Kaichen Zhang, Chunguang Zhang, Shihui Chen, Shichang Fu, Chunlei Yang, Hongyue Ma.

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
