## [Decision Letter · Decision Letter 0]

2 Dec 2025

PONE-D-25-41682PufCB-Auth: A Lightweight Multi-Factor Continuous Authentication Scheme Integrated PUF with Charging Behavior Features for EV-ChaoJi Charging Piles Interaction ScenariosPLOS ONE

Dear Dr. Ma,

Thank you for submitting your manuscript to PLOS ONE. After careful consideration, we feel that it has merit but does not fully meet PLOS ONE’s publication criteria as it currently stands. Therefore, we invite you to submit a revised version of the manuscript that addresses the points raised during the review process.

**ACADEMIC EDITOR:** 

The reviewers recommend reconsideration the manuscript with revision and modification. I invite the authors to resubmit the manuscript after addressing the comments raised by the reviewers.

I request the authors not to cite irrelevant references suggested by the reviewers

If applicable, we recommend that you deposit your laboratory protocols in protocols.io to enhance the reproducibility of your results. Protocols.io assigns your protocol its own identifier (DOI) so that it can be cited independently in the future. For instructions see: https://journals.plos.org/plosone/s/submission-guidelines#loc-laboratory-protocols. Additionally, PLOS ONE offers an option for publishing peer-reviewed Lab Protocol articles, which describe protocols hosted on protocols.io. Read more information on sharing protocols at . Additionally, PLOS ONE offers an option for publishing peer-reviewed Lab Protocol articles, which describe protocols hosted on protocols.io. Read more information on sharing protocols at https://plos.org/protocols?utm_medium=editorial-email&utm_source=authorletters&utm_campaign=protocols..

We look forward to receiving your revised manuscript.

Kind regards,

Dhanamjayulu C, Ph.D & Post.Doc

Academic Editor

PLOS ONE

“This research is funded by State Grid Corporation of China (Item No: 5400-202440202A-1-1-ZN).”

“This research is funded by State Grid Corporation of China (Item No: 5400-202440202A-1-1-ZN).”

“This research is funded by State Grid Corporation of China (Item No: 5400-202440202A-1-1-ZN).”

5. In the online submission form, you indicated that your data is available only on request from a third party. Please note that your Data Availability Statement is currently missing  contact details for the third party, such as an email address or a link to where data requests can be made. Please update your statement with the missing information.

6. Thank you for stating the following in the Financial Disclosure section:

“This research is funded by State Grid Corporation of China (Item No: 5400-202440202A-1-1-ZN).”

We note that one or more of the authors are employed by a commercial company: name of commercial company.

Additional Editor Comments :

The reviewers recommend reconsideration the manuscript with revision and modification. I invite the authors to resubmit the manuscript after addressing the comments raised by the reviewers.

I request the authors not to cite irrelevant references suggested by the reviewers

Reviewers' comments:

Reviewer's Responses to Questions

**Comments to the Author**

1. Is the manuscript technically sound, and do the data support the conclusions?

Reviewer #1: Partly

Reviewer #2: Yes

Reviewer #3: Partly

2. Has the statistical analysis been performed appropriately and rigorously? 

Reviewer #1: No

Reviewer #2: Yes

Reviewer #3: Yes

3. Have the authors made all data underlying the findings in their manuscript fully available?

Reviewer #1: No

Reviewer #2: Yes

Reviewer #3: Yes

4. Is the manuscript presented in an intelligible fashion and written in standard English?

Reviewer #1: Yes

Reviewer #2: Yes

Reviewer #3: Yes

5. Review Comments to the Author

Reviewer #1: The manuscript addresses a timely and important problem: lightweight, secure authentication for EV charging, specifically for the high-power ChaoJi standard. The proposed concept of PufCB-Auth, which fuses a hardware-level PUF with user charging behavior, is a sound and interesting approach. The manuscript is generally well-structured.

However, the manuscript suffers from several major flaws in its current form that preclude publication without significant revision.

Major Comments:

1.The experimental evaluation (Sections 4.3, 4.4) is critically insufficient. The authors only compare their proposed PufCB-Auth scheme against its own sub-components (Puf-Auth and CB-Auth). This comparison merely demonstrates that a multi-factor scheme is superior to a single-factor scheme, which is a well-established and trivial conclusion. To demonstrate the novelty and practical advantage of their specific method, the authors must compare PufCB-Auth against at least one or two existing state-of-the-art lightweight MFA schemes from the literature designed for EV or related IoT scenarios.

2.The manuscript lacks a systematic threat model and any formal security analysis. This is a critical omission for a paper proposing a new authentication protocol. While replay and modeling attacks are briefly mentioned, a comprehensive analysis of the protocol's resilience against a standard set of attacks (e.g., Man-in-the-Middle attacks, impersonation attacks, sensor capture attacks, privacy leakage of behavioral data) is entirely missing.

3.The "Enhanced PufCB-Auth" scheme described in Section 3.4 contains a severe security vulnerability. The mechanism is designed to update the digital fingerprint when an anomaly is detected. However, as the authors' own experiments show, malicious attacks (e.g., power theft, location forgery) also trigger this condition. This creates a critical flaw: an attacker can deliberately perform malicious actions to trigger a "legitimate" fingerprint update. The new fingerprint, dfnew, would then be generated based on the "most recent S charging records", which would include the attacker's poisoned data. This could be exploited for a Denial of Service (DoS) attack or to register the attacker's malicious behavior as the new "legitimate" baseline. The update mechanism must be fundamentally redesigned (e.g., to block and require out-of-band re-authentication on high-risk anomalies, not to update the profile).

4.The CB-Auth module relies on a 3σrule for its dynamic threshold (Eq. 10), which is explicitly based on the "assuming that it conforms to a normal distribution". This assumption is highly questionable for user charging behavior (e.g., location, time) which is far more likely to be multi-modal (e.g., home and work) or heavily skewed. This flawed assumption likely explains the high user-dependent variability seen in the results (e.g., the poor performance for User C vs. User E). The authors must either rigorously validate this assumption or, preferably, replace the 3σ rule with a more robust, non-parametric anomaly detection method (e.g., based on percentiles or IQR).

Minor Comments:

As stated in Q3, the experiments rely on simulated data. The authors should be required to provide the simulation scripts and parameters to ensure reproducibility, as the raw data is (justifiably) restricted.

The manuscript uses several "magic numbers" without justification or sensitivity analysis (e.g., PUF similarity threshold τ= 0.95, behavior window size S = 100, and risk threshold 0.3τΔ ). These must be justified.

Figures 3 and 4 are of poor resolution. The X-axis labels are inconsistent, unprofessional, and difficult to read. These figures must be redrawn.

A clear copy-paste error exists in the caption for Figure 3, which is incorrectly labeled as "Figure 2".

Reviewer #2: This paper proposes PufCB-Auth, a lightweight multi-factor continuous authentication scheme for Electric Vehicle (EV) charging scenarios, specifically targeting high-power ChaoJi charging piles. The method integrates two factors: a hardware-based Physical Unclonable Function (PUF) from the EV's Battery Management System and dynamic Charging Behavior (CB) features (e.g., spatio-temporal patterns, demand, frequency). These are fused into a single, compact digital fingerprint for efficient authentication. The PUF provides a unique, tamper-resistant device identity via a challenge-response protocol, while behavioral analysis detects anomalies in real-time user charging patterns. To address long-term drift in both PUF responses (due to environmental factors) and user behavior, an Enhanced PufCB-Auth version includes a dynamic fingerprint update mechanism. This update is triggered by time intervals, significant behavioral deviations, or PUF response drift. Experimental results on a Raspberry Pi-based hardware prototype demonstrate that the scheme is highly effective. Revision should be done for this version of the paper as follows:

*The motivation of the proposed method should be stated in the introduction.

* Some references are missing. Additionally, many important recent references that could support the ideas presented in this paper are also absent. The following references can help improve the content:

1- L2AI: lightweight three-factor authentication and authorization in an IoMT blockchain-based environment with unsecure channel communication. Cluster Comput 28, 865 (2025). https://doi.org/10.1007/s10586-025-05569-6

2- (2024). A lightweight hierarchical method for improving security in the internet of things using fuzzy logic. Concurrency and Computation: Practice and Experience, 36(6), e7959, https://doi.org/10.1002/cpe.7959

3- A secure and energy-efficient architecture in Internet of Things–cloud computing network by enhancing and combining three cryptographic techniques via defining new features, areas, and entities. J Supercomput 81, 944 (2025). https://doi.org/10.1007/s11227-025-07390-9

4- An authentication mechanism based on blockchain for IoT environment. Cluster Comput 27, 13239–13255 (2024). https://doi.org/10.1007/s10586-024-04565-6

* How are the specific weights (ω_t, ω_L, etc.) for the different behavioral features in the anomaly score (Equation 9) determined? Are they fixed, user-specific, or dynamically adjusted?

* The paper mentions using a "random salt" and "dynamic challenge rotation" to prevent machine learning attacks on the PUF. How is the size of the challenge pool determined, and is there a risk of exhausting it over the vehicle's lifetime?

* The update mechanism is triggered by a high anomaly score. Could this be exploited by an attacker to force frequent, unnecessary updates, potentially leading to a Denial-of-Service (DoS) by overwhelming the system?

* How does the scheme handle the initial "cold start" problem for new EV users who have little to no historical charging data to establish a reliable behavioral baseline?

* The communication overhead is calculated for authentication, but what is the total overhead during the initial registration phase of a new EV with the charging infrastructure?

* The feature update process involves generating a new fingerprint. How is the integrity and authenticity of this update process ensured to prevent a man-in-the-middle attacker from injecting a malicious new fingerprint?

* The term "PUF" is well-understood, but the paper does not specify the type of PUF being simulated (e.g., SRAM PUF, Arbiter PUF, Ring Oscillator PUF). A brief description of the PUF type emulated on the Raspberry Pi platform would add technical depth and clarify the experimental setup's realism and potential limitations.

* The paper lists various attacks but could benefit from a more structured threat model section. Explicitly stating the assumed adversary's capabilities (e.g., can they physically tamper with the BMS? Can they observe network traffic?) would better frame the security claims and highlight which specific threats PufCB-Auth is designed to mitigate.

Reviewer #3: This manuscript proposes a Lightweight Multi-Factor Continuous Authentication Scheme Integrated PUF with Charging Behavior Features for EV-ChaoJi Charging Piles Interaction Scenarios. This scheme targets electric vehicle charging pile interaction scenarios, integrating Physical Unclonable Functions (PUF) with dynamic charging behavior characteristics to establish a multi-factor continuous authentication scheme based on digital fingerprints, demonstrating considerable practical significance.

I have the following comments:

1) The title is too long; it is recommended to revise it.

2) If a table can be used to present the technologies, advantages, and disadvantages adopted by existing solutions in related works, it will be more intuitive.

3) The manuscript has conducted extensive analysis on the execution performance of the proposed scheme, but lacks a security assessment of the proposed multi-factor authentication scheme.

4) Please provide a comparative evaluation between the proposed solution and existing related solutions in terms of security and execution performance, so that the conclusion can be more convincing.

6. PLOS authors have the option to publish the peer review history of their article (what does this mean?). If published, this will include your full peer review and any attached files.). If published, this will include your full peer review and any attached files.

.

Reviewer #1: No

Reviewer #2: No

Reviewer #3: No

---

## [Author Response · Author response to Decision Letter 1]

5 Feb 2026

In addition to addressing all reviewer and editor comments in the uploaded “Response to Reviewers” document, we would like to clarify and apologize for the following administrative issues.

We apologize for the inconvenience caused by the incomplete administrative information at this stage. As noted in our previous request for a deadline extension, the revision period was relatively time-constrained, and despite our best efforts, some information could not be fully finalized in time for this submission.

Specifically, the finalized funding details are still under internal administrative confirmation, and the data-owning organization is in the process of designating an official contact for data access requests. In addition, there has been some update in the author list; however, the corresponding contact information has not yet been fully collected, and therefore the author details have not been updated in the submission system at this stage.

We sincerely apologize for these temporary omissions and will provide the complete and finalized information as soon as it becomes available, and prior to publication, in full compliance with PLOS ONE policies.

All scientific and technical comments from the reviewers and the Academic Editor have been fully addressed in the revised manuscript.

---

## [Decision Letter · Decision Letter 1]

23 Feb 2026

PufCB-Auth: A Lightweight Continuous Multi-Factor Authentication Scheme Integrated PUF with Charging Behavior Features for EV Charging

PONE-D-25-41682R1

Dear Dr. Ji Ma,

We’re pleased to inform you that your manuscript has been judged scientifically suitable for publication and will be formally accepted for publication once it meets all outstanding technical requirements.

An invoice will be generated when your article is formally accepted. Please note, if your institution has a publishing partnership with PLOS and your article meets the relevant criteria, all or part of your publication costs will be covered. Please make sure your user information is up-to-date by logging into Editorial Manager at Editorial Manager® and clicking the ‘Update My Information' link at the top of the page. For questions related to billing, please contact  and clicking the ‘Update My Information' link at the top of the page. For questions related to billing, please contact billing support..

Kind regards,

Dhanamjayulu C, Ph.D & Post.Doc

Academic Editor

PLOS One

Additional Editor Comments (optional):

The authors have addressed the reviewers’ comments properly

The article can be accepted for the publication in present form

Reviewers' comments:

Reviewer's Responses to Questions

**Comments to the Author**

1. If the authors have adequately addressed your comments raised in a previous round of review and you feel that this manuscript is now acceptable for publication, you may indicate that here to bypass the “Comments to the Author” section, enter your conflict of interest statement in the “Confidential to Editor” section, and submit your "Accept" recommendation.

Reviewer #1: All comments have been addressed

Reviewer #3: All comments have been addressed

2. Is the manuscript technically sound, and do the data support the conclusions?

Reviewer #1: Yes

Reviewer #3: Yes

3. Has the statistical analysis been performed appropriately and rigorously? 

Reviewer #1: Yes

Reviewer #3: Yes

4. Have the authors made all data underlying the findings in their manuscript fully available?

Reviewer #1: No

Reviewer #3: Yes

5. Is the manuscript presented in an intelligible fashion and written in standard English?

Reviewer #1: Yes

Reviewer #3: Yes

6. Review Comments to the Author

Reviewer #1: The authors have submitted a substantially revised manuscript that thoroughly addresses the concerns raised in the previous review round. The significant changes made to the experimental evaluation, security analysis, and protocol design have greatly improved the technical soundness of the work.

Specific improvements include:

Comparative Evaluation: The inclusion of external baselines—specifically PUF-MAP (hardware-centric) and L2AI (behavior-centric)—strengthens the evaluation considerably. The new data in Tables 1–4 clearly demonstrates the proposed scheme's advantages regarding communication overhead and storage efficiency compared to state-of-the-art alternatives.

Security Vulnerability Mitigation: The redesign of the "Enhanced PufCB-Auth" update mechanism effectively addresses the previously identified poisoning vulnerability. By strictly decoupling risk assessment from baseline updates and ensuring that only authentication-passed records are used for model retraining, the system is now resilient to the forced-update attacks mentioned in the previous review.

Threat Model: The addition of a formal "Threat Model and Security Evaluation Methodology" (Section 5.4) provides the necessary context for the security claims and allows for a structured assessment of the protocol's resilience.

Statistical Validity: The clarification regarding the statistical assumptions in the CB-Auth module—specifically that the adaptive thresholding applies to the aggregated deviation score rather than raw behavioral features—is technically acceptable for this context.

Reproducibility: The addition of Appendix B, which details the simulation parameters and data generation methods, significantly improves the reproducibility of the experiments.

The manuscript is now technically sound and makes a valid contribution to the field of EV charging security.

Reviewer #3: In the first round, I raised some questions, and the authors have made revisions and provided responses accordingly. I believe the manuscript is now acceptable.

7. PLOS authors have the option to publish the peer review history of their article (what does this mean?). If published, this will include your full peer review and any attached files.). If published, this will include your full peer review and any attached files.

.

Reviewer #1: No

Reviewer #3: No

---

## [Editor Report · Acceptance letter]

PONE-D-25-41682R1

PLOS One

Dear Dr. Ma,

I'm pleased to inform you that your manuscript has been deemed suitable for publication in PLOS One. Congratulations! Your manuscript is now being handed over to our production team.

Kind regards,

on behalf of

Dr. Dhanamjayulu C

Academic Editor

PLOS One